# Reliability of Family Dogs’ Sleep Structure Scoring Based on Manual and Automated Sleep Stage Identification

**DOI:** 10.3390/ani10060927

**Published:** 2020-05-24

**Authors:** Anna Gergely, Orsolya Kiss, Vivien Reicher, Ivaylo Iotchev, Enikő Kovács, Ferenc Gombos, András Benczúr, Ágoston Galambos, József Topál, Anna Kis

**Affiliations:** 1Institute of Cognitive Neuroscience and Psychology, Research Centre for Natural Sciences, 1117 Budapest, Hungary; kisorsolia@gmail.com (O.K.); eniko.k.0531@gmail.com (E.K.); agoston.galambos89@gmail.com (Á.G.); topal.jozsef@ttk.mta.hu (J.T.); vargane.kis.anna@ttk.mta.hu (A.K.); 2Department of Ethology, Eötvös Loránd University, 1117 Budapest, Hungary; vivien.reicher@gmail.com (V.R.); ivaylo.iotchev@gmail.com (I.I.); 3Department of General Psychology, Pázmány Péter Catholic University, 1088 Budapest, Hungary; gombos.ferenc@btk.ppke.hu; 4Institute for Computer Science and Control, Informatics Laboratory, 1111 Budapest, Hungary; benczur.andras@sztaki.hu; 5Department of Cognitive Psychology, Eötvös Loránd University, 1053 Budapest, Hungary

**Keywords:** canine EEG, sleep staging, polysomnography reliability, automatic staging

## Abstract

**Simple Summary:**

Sleep alterations are known to be severe accompanying symptoms of many human psychiatric conditions, and validated clinical protocols are in place for their diagnosis and treatment. However, sleep monitoring is not yet part of standard veterinary practice, and the possible importance of sleep-related physiological alterations for certain behavioral problems in pets is seriously understudied. Recently, a non-invasive electroencephalography (EEG) method was developed for pet dogs that is well-suited for untrained individuals. This so called polysomnography protocol could easily be implemented in veterinary diagnosis. However, in order to make the procedure more effective and standardized, methodological questions about the validity and reliability of data processing need to be answered. As a first step, the present study tests the effect of several factors on the manual scoring of the different sleep stages (a standard procedure adopted from human studies). Scoring the same recordings but varying the number of EEG channels visible to the scorer (emulating the difference between single channel versus four channel recordings) resulted in significant differences in hypnograms. This finding suggests that using more recording electrodes may provide a more complete picture of dog brain electrophysiological activity. Visual sleep staging by three different expert raters also did not provide a full agreement, but despite this, there were no significant differences between raters in important output values such as sleep structure and the spectral features of the EEG. This suggests that the non-invasive canine polysomnography method is ready to be implemented for veterinary use, but there is room for further refinement of the data processing. Here, we describe which parts of the sleep recording yield the lowest agreement and present the first form of an automated method that can reliably distinguish awake from sleep stages and could thus accelerate the time-consuming manual data processing. The translation of the findings into clinical practice will open the door to the more effective diagnosis and treatment of disorders with sleep-related implications.

**Abstract:**

Non-invasive polysomnography recording on dogs has been claimed to produce data comparable to those for humans regarding sleep macrostructure, EEG spectra and sleep spindles. While functional parallels have been described relating to both affective (e.g., emotion processing) and cognitive (e.g., memory consolidation) domains, methodologically relevant questions about the reliability of sleep stage scoring still need to be addressed. In Study 1, we analyzed the effects of different coders and different numbers of visible EEG channels on the visual scoring of the same polysomnography recordings. The lowest agreement was found between independent coders with different scoring experience using full (3 h-long) recordings of the whole dataset, and the highest agreement within-coder, using only a fraction of the original dataset (randomly selected 100 epochs (i.e., 100 × 20 s long segments)). The identification of drowsiness was found to be the least reliable, while that of non-REM (rapid eye movement, NREM) was the most reliable. Disagreements resulted in no or only moderate differences in macrostructural and spectral variables. Study 2 targeted the task of automated sleep EEG time series classification. Supervised machine learning (ML) models were used to help the manual annotation process by reliably predicting if the dog was sleeping or awake. Logistic regression models (LogREG), gradient boosted trees (GBT) and convolutional neural networks (CNN) were set up and trained for sleep state prediction from already collected and manually annotated EEG data. The evaluation of the individual models suggests that their combination results in the best performance: ~0.9 AUC test scores.

## 1. Introduction

Family dogs are now a widely accepted model of human socio-cognitive functioning [1], and this role was recently extended to neurocognition, including sleep physiology [2]. Due to ethical considerations, a non-invasive polysomnography (PSG) method had been developed for studying different aspects of sleep in pet dogs (see [3]). This method has several advantages: (1) it can be easily applied without any previous training of the subjects, (2) it causes no pain or serious stress to the animal, (3) it provides comparable macrostructural and spectral results to data from both human and mammalian PSG studies, and (4) electrode placement and data acquisition are easy to learn and carry out by following the validated protocol (cf. [3,4]). 

While dog PSG studies are relatively easy to carry out, the field of canine sleep research faces similar practical challenges regarding data processing as human studies do. The most time-consuming part of the process is manual sleep staging (i.e., identifying the sleep stage(s) in segments of an EEG-recorded signal) and artifact rejection (i.e., eliminating artifacts resulting from muscle tone or eye movements). Manual/visual scoring is still the gold standard in human sleep studies as well, wherein a human scorer codes the PSG recordings epoch-by-epoch (usually in 20–30 s intervals) and identifies the sleep stage according to standard criteria [5]. Several attempts have been made to automate this part of the data processing and have the (human) PSG recordings scored by a computer program (e.g., [6,7]).

It would reduce the cost and improve reproducibility if the automated sleep stage scoring algorithms achieved comparable accuracy to manual scoring. Even the use of semi-automatic scoring, in which the sleep–wake distinctions are made automatically and provide some quality control over the procedure, would also be beneficial and would make this time-consuming process easier. In human PSG data, there are some promising results regarding computer-assisted or automated staging technologies [8,9,10] using a wide range of signal processing methods and algorithms (for a review, see [11]).

Most of the studies focusing on the development of automated sleep stage scoring algorithms are in need of feature selection since they are not capable of running on the raw dataset. Feature generation is usually the second step after the sufficient cleaning and processing of the raw dataset; it reduces dimensionality in the dataset, which improves the speed and performance. It is an important task for several supervised machine learning models, especially in time series data feature generation [12]. These approaches first define the most common features such as frequency, amplitude or other time-based features of EEG waveforms. In the case of neural networks, raw data are used as inputs and feature selection is included in the algorithm. For other algorithms, feature selection/feature generation is usually applied for input-preprocessing after data-cleaning. [13,14]. However, there is no consensus in the literature about the training process or the feature generation, and most of the studies rely on prior knowledge to compute representative features to effectively characterize EEGs [9,15]. These prevent the generalizability to larger datasets from various patients with different sleep patterns, and the applicability is even more limited in non-human research.

Several technical and biological artifacts, such as active power line, impedance fluctuation, eyeblink, and muscle activity might interfere with EEG signals. Similarly to sleep staging, artifact rejection can be done manually or automatically, usually by independent component analysis. Some types of artifact cannot be easily eliminated with the application of preprocessing algorithms. Therefore, there is still no consensus about the efficiency of fully automatic artifact rejection (e.g., [16]). 

It is customary in both human and dog PSG studies to report an inter-rater reliability (IRR) measurement (e.g., agreement rate (%) or Cohen’s kappa (κ)) to prove that sleep stage scoring has been done by expert coders who highly agree in their assessments (for a review, see [17]). Such IRR scores are usually based on a randomly selected sample of the whole dataset (i.e., 2%–20% of the whole dataset). However, some studies on human sleep, that aimed to explore the reliability of visual scoring, calculated IRR based on whole dataset (for a review, see [17]). In general, the IRR score of human PSG studies depends largely on the sleep stage (e.g., [18]), the rater’s scoring experience [19] and the number of active electrodes placed during the recording [20], and there is considerable variability between the individual records being scored [21]. The average overall IRR (κ value) is usually 0.6–0.8 (for a review, see [17]); however, much lower agreements (0.2–0.4) can be found in several studies, especially in the NREM1 phase, (non-rapid eye movement), while for wake and REM phases, the agreement is usually higher (0.8–0.9) [18,20]. Danker-Hopfe and colleagues investigated the IRR between scorers from different laboratories and found similar κ values as in previous studies; however, they also studied the macrostructural pattern of the hypnograms [22]. They showed that despite a relatively low κ, which was found in the case of the NREM1 and NREM2 phases, the intraclass correlation of macrostructural variables (e.g., wake after sleep onset (WASO), sleep onset latency and total sleep time) was still very high (>0.86). This suggests that sleep macrostructure can be reliable even with a relatively low IRR.

Dog PSG studies so far have been conducted on relatively low samples sizes (between *N* = 7 and *N* = 16 individuals), and such datasets are typically scored by one coder (e.g., [2,23]). The reported inter-rater agreements are high (0.8–0.9), but have been performed on a very low number of randomly selected epochs. More recently, however, a dataset of over 150 dogs was used in one study [24], with the recordings being analyzed by several different scorers with different scoring experience. There is, thus, a rising importance of carrying out in-depth analyses of within- and between-coder reliability for canine PSG scoring.

A further factor to be considered is that our canine PSG method evolved from its initial single-channel form (as described originally in [3]) to a four channels protocol. In addition to Fz, channel Cz, F7 and F8 are also included (for a detailed description of the updated procedure, see [24]). This four channels protocol not only allows for topographical comparisons [24] but also provides more data for sleep stage identification. In humans, it has been shown that hypnogram scoring based on single-channel EEG, contrary to a full PSG protocol, yields different results [20], thus the same question is highly relevant for canine sleep studies as well.

The aim of the present study is to provide a detailed descriptive analysis of reliability measures of dog sleep staging based on the non-invasive canine PSG method. Study 1 will focus on manual scoring by providing inter-rater agreement measures and error matrices for full 3 h-long recordings between coders with different scoring experience, as well as examining the possible effects of the number of visible channels used when scoring (within-coder). Importantly, the effects of any sleep stage scoring differences on the macrostructural and spectral results will also be reported. Study 2 will present the results of the first version of automatic scoring (wake versus sleep identification) of canine PSG recordings using machine learning algorithms.

## 2. Study 1

### 2.1. Materials and Methods

#### 2.1.1. EEG Recordings

In order to investigate inter-scorer reliability, 10 PSG recordings (of 10 different dogs) were randomly chosen from the Family Dog Project database (6 males, 4 females, mean age 6.9 ± 4.2 years; 2 Golden Retrievers, 1 Labrador Retriever, 1 Cocker Spaniel, 1 Springer Spaniel, 1 Mudi and 4 mongrels). Recordings were carried out in accordance with the guidelines for the use of animals in research described by the Association for the Study of Animal Behaviour (ASAB); Ethical Committee statement number: PE/EA/853–2/2016. All recordings contained a 3 h-long afternoon sleep recording (first sleeping occasion of the dogs without any pre-sleep treatment or stimuli), which took place in the sleep laboratory of the Research Centre for Natural Sciences, Budapest. Surface-attached scalp electrodes were placed over the anteroposterior midline of the skull (Fz, Cz, G2) and on the zygomatic arch (*os zygomaticum*), next to the right eye (F7) and left eye (F8). The ground electrode was placed on the left musculus temporalis (G1) (see Figure 1). Electrodes were placed bilaterally on the musculus iliocostalis dorsi for electromyography (EMG) and over the second rib for electrocardiography (ECG). Respiratory movements were also monitored by a respiratory belt attached to the chest (PNG). Before attaching each electrode, we used PARKER^®^ SIGNA^®^ spray to clean the surface of the skin. Gold-coated Ag|AgCl electrodes, fixed with EC2 Grass Electrode Cream (Grass Technologies, Warwick, RI, USA), were used for the recordings. The impedances for the EEG electrodes were kept below 20 kΩ. The Fz, Cz, F7 and F8 derivations served as the EEG signal. Signals were collected, pre-filtered, amplified and digitized at a sampling rate of 1024 Hz/channel, by using the 25 channel SAM 25R EEG System (Micromed, Mogliano Veneto, Italy), as well as the System Plus Evolution software, with second-order filters at 0.016 Hz (high pass) and 70 Hz (low pass).

#### 2.1.2. Coders and Visual Scoring

For the sleep stage scoring, custom-made software was used (Fercio’s EEG Plus^©^ Ferenc Gombos 2009–2020, Budapest, Hungary). The scoring (as described in [3]) identified the following stages: wakefulness (W), drowsiness (D), non-REM (NREM) and REM sleep (see Table 1 for detailed scoring criteria). Three coders were involved in the visual scoring procedure, two of whom were the researchers originally recording and scoring the measurements for previously published experiments (hereafter Coders 1A and 1B), and the third coder (hereafter Coder 2) carried out the reliability coding for the purpose of the current study. Coder 1A (V Reicher), the most experienced rater who had already scored more than 200 canine PSG recordings in the past, scored four out of the ten recordings (originally included in [25]). Coder 1B (I Iotchev), the least experienced rater, who had scored approximately 50 PSG recordings so far, scored six out of the ten recordings (originally included in [26]). Coder 2 (A Gergely), who was in between Coder 1A and 1B regarding expertise, had scored approximately 110 canine PSG recordings in the past and coded all ten recordings twice (both with one and four EEG channels visible). In the end, the full duration of each PSG recording was scored for a total of three times: Coder 1A (*N* = 4) and Coder 1B (*N* = 6) scored the recordings with all 4 EEG channels visible (Fz, Cz, F7 and F8; in addition to the ECG, EMG and respiration channel), and Coder 2 (A Gergely) scored all 10 PSG recordings with the same settings (4 EEG channels visible), as well as with only 1 EEG channel visible (Fz; in addition to the ECG, EMG and respiration channel). The two independent codings of Coder 2 were carried out in a counterbalanced order with a retention interval of 7 days. All sleep stage scorings were blind to subject details.

The three independently scored hypnograms resulted in one sleep macrostructure dataset each. The following macrostructural variables were exported: sleep latency 1 (first epoch, i.e., 20 s long segment, scored as D sleep (= drowsiness) from recording onset, min), sleep latency 2 (first epoch scored as NREM sleep from recording onset, min), relative wake duration (%), relative D duration (%), relative NREM duration (%), relative REM duration (%), wake after sleep onset (WASO, time spent awake after first epoch scored as D, min), and average sleep cycle duration (average duration of an uninterrupted D + NREM + REM phase, min) [3].

#### 2.1.3. Artifact Rejection and EEG Spectra

Artifact rejection was carried out manually by Coders 1A and 1B, respectively, on 4 s-long epochs before further automatic analysis. Then, relative EEG power spectral values of the Fz channel were obtained for each 0.25 Hz frequency bin (see [3]) for D, NREM and REM stages separately as a proportion of total (1–30 Hz) power. These relative spectra were calculated separately for the three hypnograms scored by Coder 1 (A + B) and Coder 2 (1- and 4 channels scoring) using the same artifact file.

#### 2.1.4. Data Analysis

Inter-rater reliability (agreement rate (%) and Cohen’s kappa (κ)) was calculated in order to examine agreements in visual scoring between coders and between one and four channel scoring (within-coder). To do so, the total numbers of agreements (i.e., number of epochs with an identical score) and disagreements (i.e., number of epochs scored differently) were calculated between Coder 1A, Coder 1B and Coder 2, as well as between 1 and 4 channel scoring by Coder 2. The total number of epochs scored as W, D, NREM and REM by Coder 1A and 1B, Coder 2—1 channel scoring and Coder 2—4 channel scoring were used to calculate the κ values. A κ value of 0–0.2 is considered slight agreement; 0.21–0.4, fair agreement; 0.41–0.6, moderate agreement; 0.61–0.8, substantial agreement; and >0.8, almost perfect agreement [27].

Based on the visual inspection of the error pattern (see Results), we proceeded to examine whether inter-rater reliability can be increased by eliminating 60 s (three epochs) before and after every sleep stage change (where most disagreements seemed to be located). Thus, κ was calculated in the same way as described above for this reduced and “corrected” dataset as well. As a reference, we used the four channel scoring of Coder 2 (i.e., changes in the sleep stages in this output served as the reference for cutting the 60 s out from Coder 2’s 1 channel scoring).

Previously published studies on canine EEG usually calculated inter-rater reliability on randomly selected parts of the full dataset (e.g., [3,4,23]). In order to see if there is any difference in κ calculated on the full and on randomly selected parts of the dataset, we also calculated κ for 20 randomly selected epochs in five randomly selected subjects (i.e., 100 epochs in total).

Confusion matrices were also calculated both for within- and between-coder conditions in order to get a descriptive picture of the effect of visible channels and different coders on the visual scoring (separately for W, D, NREM and REM stages).

Paired t-tests were applied to compare the exported macrostructural variables between coders, and between 1 and 4 channel scoring in the within-coder condition.

The relative spectral power of the Fz channel for all sleep stages (D, NREM and REM) was compared between Coder 1 (A + B) and Coder 2, as well as between 1 and 4 channel scoring by Coder 2, with paired t-tests, separately for the four frequency ranges of delta (1–4 Hz), theta (4–8 Hz), alpha (8–12 Hz) and beta (12–30 Hz).

An adjusted *p*-value was used for multiple comparisons (Bonferroni correction, adjusted *p* = 0.006), and SPSS ver. 22, for statistical analysis.

### 2.2. Results and Discussion

The agreement rate (%) and Cohen’s kappa (κ) values for the different inter-rater comparisons are shown in Table 2 in ascending order. The κ values proved to be considerably lower than those reported in our previous studies that applied the same non-invasive canine PSG. Importantly, however, the κ value calculation in the previous studies was based on very few epochs (e.g., 10 randomly chosen epochs per recording in [3], thus accounting for only 0.4%–2.3% of the whole dataset). While in these previous studies, the Cohen’s κ was 0.9–0.98, which is considered an almost perfect agreement [3,4,23], in the present study, only a substantial agreement (0.73) was found even as the most favorable outcome (within-coder for randomly selected 100 epochs). It should also be noted that the number of visible channels was different (one vs. four) during the two scorings. An additional important finding is that a remarkable difference can be seen between the κ values when the whole dataset vs. 100 epochs were used for inter-observer agreement calculation (see Table 2).

According to scorers’ previous experience, we found that Coder 2 had a higher agreement rate with the more experienced Coder 1A than with the less experienced Coder 1B. This highlights the importance of scoring experience and more controlled, supervised and standardized scoring methodology for canine PSG.

In human EEG studies, the inter-rater reliability between different coders (if any) is usually calculated on the basis of the full dataset (e.g., [22,28]), and the κ value is generally between 0.6 and 0.8 depending on the sleep stage (for a review, see [17]). Regarding the number of visible channels, in human sleep literature, there is evidence that sleep staging is more challenging when rating single-channel vs. PSG records, and a 0.67 κ value was shown when comparing the two [20]. In our study (using one versus four EEG channels as opposed to the 19 EEG channels in the standard 10–20 PSG protocol used in human studies), a very similar κ value was found when we eliminated 60 s from the point of transition between sleep stages from the data (see Table 2 for details). Thus, the elimination of some epochs right before and after sleep stage transitions somewhat increased the reliability, reaching a κ value similar to that in human sleep literature. Interestingly, the elimination of epochs before and after transitions was not as effective in the between- as in the within-coder condition to increase IRR.

The confusion matrix shows the ratio of epochs scored as W, D, NREM and REM by Coder 2 based on one channel and four channel scorings (Table 3) as well as Coder 2 vs. Coder 1A (Table 4) and Coder 1B (Table 5). As expected, the highest ratios of epochs were coded identically in all sleep stages, except drowsiness in the Coder 1B vs. Coder 2 condition (Table 5). In accordance with what is shown by the κ values, the ratio of agreement is higher in the within- than in the between-coder condition and also shows higher agreement between Coder 2 and Coder 1A than Coder 1B. Similarly to human studies, the level of agreement is highly dependent on sleep stage in each condition (for a review, see [17]). Interestingly, the wake and NREM stages are scored similarly, with the highest ratios of agreement irrespective of condition, while D and REM reached low agreement. In the within-coder condition the most common disagreement was due to D, as epochs scored as D during four channel scoring were frequently scored as W, NREM or REM during one channel scoring. A similar tendency can be found in between-coder conditions, where Coder 2 frequently scored an epoch as W, while Coder 1B had already scored it as D. It also occurred quite frequently that epochs scored as NREM by Coder 1B were still coded as D by Coder 2. According to the human sleep literature, REM sleep shows the highest reliability, which is not surprising, since the REM phase is the most studied sleep stage; therefore, a great amount of effort has been made in the description and discrimination of REM sleep [19]. At the same time, NREM sleep, which is the most diverse and complex phase in human sleep, especially Stage S1/(N1) (partly overlapping with drowsiness in dogs [3]), has the lowest inter-scorer agreement (e.g., see [17,19,20,29] for a review). The high agreement on the NREM phase in the present study might be due to the fact that contrary to human sleep, canine NREM sleep has not been divided into different stages; at the same time, high voltage (>75 µV) and delta activity can be found only during NREM.

Taken together, these results show that, just like in the visual scoring of human sleep stages [17], drowsiness is the most difficult to detect in dogs. We assume that the relatively low agreement on REM in dogs is because, contrary to human subjects, dogs are characterized by polyphasic sleep and thus at the end of the NREM phase, dogs equally often wake up as they switch to REM. This—together with the fact that dogs’ sleep structure, in general, contains more stage shifts than humans’, which we have shown here to be the most critical points for disagreement—explains the slightly lower inter-rater agreement.

A pairwise comparison of the sleep macrostructure values obtained from the three hypnograms (Table 6) showed no differences between one and four channel (within-coder) scorings (all *p* > 0.1). In the case of the between-coder comparison of sleep latencies, the relative wake and NREM durations did not differ (*p* < 0.08) after correcting for multiple comparisons (adjusted *p* = 0.006). The mean (± SE) values for sleep macrostructure obtained from the three different scorings are shown in Figure 2. Illustrative hypnogram examples to show individual-level scoring differences are presented in Figure 3.

Regarding the within- and between-coder condition, paired t-tests on the relative spectral power of the Fz channel showed no significant differences in most of the frequency ranges (i.e., delta, theta and alpha) in any of the sleep stages (D, NREM and REM) (Table 7). However, the beta frequency of the Fz channel showed a difference between coders in D, although it did not remain significant after correction for multiple comparisons (Table 7). This means that a higher beta spectral power value was calculated in case of Coder 1A + B compared to Coder 2. This is in line with the tendency that epochs scored as D by Coder 1B were frequently scored as W by Coder 2, and thus epochs in the transition of W and D containing more beta components were included in the drowsiness spectrum of Coder 1B but excluded from Coder 2′s. It has to be noted that the present analysis focused solely on the effect of sleep stage scoring on EEG spectrum outcomes and used the same artifact file for all three datasets. Disagreements in artifact scoring (not studied here) might induce further noise in the data.

## 3. Study 2

### 3.1. Materials and Methods

All available recordings were taken from the Family Dog Project database with the aim of testing the automated sleep–wake detection algorithm. All previous PSG recordings were used where the measurement was not preceded by any pre-treatment (this included recordings reported in previous publications [3,4,23,25,30] as well as unpublished recordings, which had been scored by a total of six different scorers). A total of 304 recordings of a ca. 3 h duration each from 151 dogs (age range: 2 months–17 years, 6.44  ±  3.97 (mean  ±  SD); 79 females, 72 males; 97 neutered, 38 intact and 19 of unknown reproductive status; 96 purebred from 42 different breeds and 55 mongrels) were included in the final dataset. For all the recordings, the manually scored hypnogram was used as an annotation reference. PSG recordings in which the percentage of sleep label was less than 5% or the proportion of awake labels was less than 2% (*N*  = 3) were excluded from all analyses. Each PSG has an accompanying hypnogram containing expert annotations (visual scoring) of sleep stages. The sleep stages were scored as described in Study 1 as Wake (1), Drowsiness (2), NREM (3) and REM (4). During the modeling, the sleep stages were merged into a binary format: Sleeping (>1) and Wake (1).

The algorithms used five time series including brain activity (EEG, Fz-G2 derivation), eye movements (EOG, F7-F8 derivation), muscle activity or skeletal muscle activation (EMG), heart rhythm (ECG) and respiration (PNG), all recorded simultaneously. The signals were obtained with a sampling rate of either 249 Hz (in case of 27 recordings) or 1024 Hz. Each recording was split into 20 s long epochs, and annotations for each epoch were added to the hypnograms.

Machine learning (ML) models were trained with the intention of using input features calculated from the five time series. The entire dataset was split randomly into training and test sets. Eighty-percent of the data were used for training, and 20%, for testing our models. Note that the data were split across recordings, i.e., a single recording either belonged to the training or the test sets. This ensured avoiding the overfitting of our models upon recording specific signals. In order to prepare the data for modeling, the following processing and feature generation steps were applied:In order to reduce noise in the data, each time series was standardized and split into 100 ms intervals, and the mean value was used for each interval.Five more time series were added that contained the absolute value of the original time series, as seen in Figure 4.Each label belonged to one 20 s-long epoch. We had calculated the features for modeling based on the assumption that adjacent epochs might contain information about the current epoch. Thus, we added the 10 s-long period before the epoch, and the 10 s-long period after the epoch to the 20 s-long period of the epoch. This 40 s-long interval was used for feature generation (see Figure 4).Since the initial averaging was done per 100 milliseconds, 10 s are represented by 100 data points. Each epoch label hence corresponds to 400 data points of the five time series from the raw data plus five time series for their absolute values (see Figure 4).Finally, features were generated based on the 400 data points per time series per epoch.We split the 40 s extended epochs into time windows with a set of different lengths, *tw*, between 1 and 40. For example, in the case of *tw* = 4, we created 10 subintervals within the 40 s-long intervals belonging to the epoch.The mean, standard deviation, maximum and minimum were calculated within each time window (see Figure 5). For example, with *tw* = 4 s, descriptive statistics of the 10 subintervals were generated.

The final dataset contains 138,962 records in the training dataset and 32,716 records in the test dataset.

To evaluate the quality of the prediction on the test set, the area under the curve (AUC) was used [31]. AUC is a classification metric that shows how much the model is capable of distinguishing between classes. The higher the AUC (between 0.5 and 1), the better the model is at distinguishing between classes.

### 3.2. Modeling, Results and Discussion

The first machine learning model that we used is logistic regression (LogReg), a linear model for binary classification [32]. The model fits a linear decision boundary between the classes. The logistic regression models were trained by using the scikit-learn Python library [33] with default parameters. The feature importance values were calculated by taking the absolute values of the linear model coefficients. As indicated in Table 8, the standard deviation calculated from the absolute values of the EEG signal and the PNG are the most important. The best performing model was found when all the attributes of the 5, 10, 20 and 40 s resolutions were used. The AUC of the best model was 0.864.

Our second learning machine was gradient boosting trees (GBTs), which recently became popular for supervised learning tasks [34]. In a GBT, we are creating decision trees in an iterative way to incrementally improve its predictive performance. To reduce the risk of overfitting, a GBT classifier in this paper is implemented using the scikit-learn library using the following parameters: number of the decision trees, 200; learning rate, 0.1; maximum depth of each decision tree, 2. GBT achieved an AUC of 0.872 when trained on the same dataset as the logistic regression.

The GBT model outputs feature importance values based on the number of times it uses a given feature when building the decision trees. As we can see in Table 9, the EEG signal is also an important attribute in the GBT, but in this model, the EOG is even more important. Contrarily to logistic regression, GBT uses statistics of the original signal.

As our final experiment, convolutional neural networks (CNN) were trained on the data. Time series can be naturally handled with CNN architectures. Hence, in case of the CNN, we were using the raw time series with the 400 data points without any feature generation. We used Keras [35] to create the model and collect the experimental results, with Tensorflow [36] as a backend. Our model used a three-layer convolutional neural network with ever shorter convolutional window sizes (10, 5 and 2) in successive convolutional layers. Thus, convolutional operations were performed first in 1 s intervals and then at 5 and 10 s intervals. The convolutional operations approximate the different aggregated statistics computed for logistic regression and GBT, but here, we let the model learn the operation (mean, min, max or any function) accurately.

The model was trained with the following parameters: kernel-sizes of the convolutional layers, 10, 5, 2; number of iterations, 9; batch size, 64; starting learning rate, 0.01; optimizer, Adam. The AUC of the CNN was 0.880.

Finally, the predictions of the above three models were averaged to gain the best predictive performance. The combination of the LogReg, GBT and the CNN performed better than the individual models and resulted in a final 0.890 AUC score.

## 4. Discussion

The present study provided reliability measures for both manual and automated sleep stage scoring in non-invasively recorded dog PSG. Previous studies using the protocol for canine PSG by Kis et al. [3] all used manual scoring to obtain hypnograms, and the present findings confirm that despite disagreements at various levels, this is a valid approach resulting in reliable macrostructural and EEG spectral data.

Automated scoring is often preferred over visual inspection due to its objective nature (e.g., arguments in [37], but see [38]). In canine sleep research, sleep spindle detection has been successfully automated in recent years [24,39]; although the method was not validated against visual detection, external validity was proven at different levels. While there is no consensus regarding sleep spindle identification among human researchers either (both automated and manual methods are in use—e.g., [37,38]), sleep stage scoring is almost exclusively carried out manually, despite continued efforts to develop a fully automated method (e.g., [7]). The machine learning algorithm presented in this paper performed relatively well for binary classification (distinction between sleep and awake), but the next step would be to explore the possibility of multiclass annotation (distinguishing the different sleep stages) by defining more elaborative features and building more accurate multiclass models. At the current stage, the model could be of practical use for accelerating data processing via a semi-automated method where sleep–wake distinctions are made automatically whereas sleep stages (D, NREM, REM) are scored manually. Given the huge individual variability that characterizes dogs at the physiological level (e.g., [40,41]), such semi-automated processes seem to be the best practice so far and have been implemented, e.g., for heart-rate analysis [41,42].

In addition, the current machine learning experiments have made little use of the fact that data consist of time series and approached the task solely as a classification problem. An interesting area of research would be to approach this machine learning task with an increased focus on time series analysis and to incorporate the regularities in the sequence of sleep stages into the algorithm. Since most dog sleep research studies apply a within-subject design (with an adaptation occasion followed by two pre-sleep treatments administered in a random order), advantage can be taken of the fact that the algorithms can be trained with the first recordings of the dogs and then be used for scoring later recordings of the same dogs.

Results from Study 1, showing that (manual) sleep staging is least reliable when transitions occur between stages, could also be taken into consideration for optimizing a semi-automated method. While cutting out the epochs before and after a transition seems to somewhat increase the reliability of the scoring, it also results in additional data loss, and this trade-off relationship should thus be further examined before implementing it into practice. In addition, considering the higher disagreement between coders with less experience, it might be optimal to train/adjust the learning algorithms separately for a given coder if a certain number of coded recordings are available. In line with our assumption, we found the highest agreement within coders despite the fact that the number of visible EEG channels differed between the two intra-observer scoring events. Presumably, this agreement would be even higher with an identical setup of visible channels during scoring. For future canine sleep studies, we therefore suggest one scorer only when coding PSG recordings for comparisons and using the same set of visible EEG channels during visible sleep staging. 

There seems to be a difference in reliability between the different sleep stages. While the most optimal automated algorithm achieved a very high (AUC > 0.88) agreement between sleep and wake, manual scoring seems to struggle most with the distinction between W and D, which is in line with reports on human data, where Stage 1 NREM sleep seems to be the most problematic with regards to scoring (for a review, see [17]). This, again, suggests that the implementation of an objective and automated method for sleep–wake distinction might cause a significant improvement in reliability. Alternatively, when in a human study authors decided to eliminate Stage 1 NREM completely from the PSG recordings, it resulted in an increased κ value [20]. Eliminating D from dogs’ hypnograms in future studies might be a feasible option but needs further investigation. The mere existence of the D stage in carnivore and insectivore species is suggestive for the lack of a clear-cut distinction between sleep and awake [43], thus a strict categorization during hypnogram scoring is likely not ideal. Furthermore, the current canine PSG protocol does not make it possible to record from occipital derivations where alpha activity is best observed in humans [20], which would be crucial for recognizing quiet W/D stages. Previous studies using the same non-invasive EEG method and hypnogram scoring protocol seem to indicate that dogs’ D reacts to pre-sleep treatments mostly in the same ways as NREM, although the results are often less clear for that stage. For example, relative D and NREM durations were both higher after positive as opposed to negative social interaction, whereas relative REM duration clearly showed the opposite trend [30]. Following a behaviorally active day, dogs showed a marked increase in NREM duration compared to that following a behaviorally passive day. Meanwhile, the same tendency was not significant for D in the two conditions [3]. According to another study with a different protocol [4], however, following an active day, dogs were more likely to have an earlier drowsiness and NREM. At the same time, dogs spent less time in D and more time in NREM and REM, compared to after a normal day. The spectral analysis of the D stage is additionally problematic due to the high proportion of muscle tone artifacts, which led to the exclusion of this stage from an analysis showing that learning new commands had an effect on the EEG spectrum in dogs [23].

Our results show that similarly to in human studies on sleep, the IRR of canine PSG analysis is highly dependent on the scorers’ prior experience. It has been proposed that ongoing training and scoring experience at the time of the experiment has the potential to reduce differences between the scorers of human PSG recordings [28]. While there are currently no standard criteria for how many PSG recordings a person needs to code in order to become a so-called expert, in this study, all three coders had coded at least 80 PSG recordings prior to the scorings included in the analysis. Some suggest, however, that this requirement may be not enough [28]. In spite of these limitations, high agreement between expert scorers was found for the identification of NREM (Table 3, Table 4 and Table 5). Some of the more recent work on dog PSG data exclusively examines NREM traces [24,26,39], while comparisons of the relative ratio of REM to NREM duration [4,30] are also relatively safe from the difficulty of distinguishing D from W. This suggests that visual scoring in the dog can be sufficient with regard to many questions.

Another possible method for obtaining better agreement in canine sleep staging would be the application of an alternative scoring system, which has been proposed by the American Academy of Sleep Medicine (AASM) [44]. Compared to the most widely accepted standard described by Rechtschaffen and Kales [5], the AASM classification comprises alternative EEG derivations, merging non-REM Stages 3 and 4 into N3, and recommends different sampling rates and filter settings for PSG reporting. The AASM system has been revised and reviewed several times and might have the potential to revolutionize visible and automated canine PSG scoring as well. Alternatively to distinct sleep-staging, researchers frequently evaluate human PSG data based on quantitative sleep parameters (amplitude- and frequency-mapping etc.) or by feature-plots (arousals, slow-wave activity, spindle-plots etc.). Considering and describing sleep as a continuum can be a feasible alternative in canine sleep research as well, especially in automated wake–sleep identification (e.g., [45]).

While there is evidence supporting the external validity of the canine PSG protocol [3], as the data obtained have revealed human-like memory consolidation and emotion processing during sleep, there is still room for methodological improvement in the field. The present study explores this space by examining the reliability of the visual sleep stage scoring applied thus far and also by offering and evaluating a first step towards a semi-automated scoring method.

## 5. Conclusions

In summary our results showed that the reliability of manual canine sleep stage scoring can be influenced by the number of visible channels during staging and the expertness (i.e., coding experience) of the raters. Most importantly we revealed no difference between coders in macrostructural and spectral variables inspite the relatively low inter-coder agreement scores, which suggest that most of the disagreements are eliminated with artefact rejection. We also made the first step toward automatic canine sleep stage scoring that reliably distinguish between sleep and awake.

## Figures and Tables

**Figure 1 animals-10-00927-f001:**
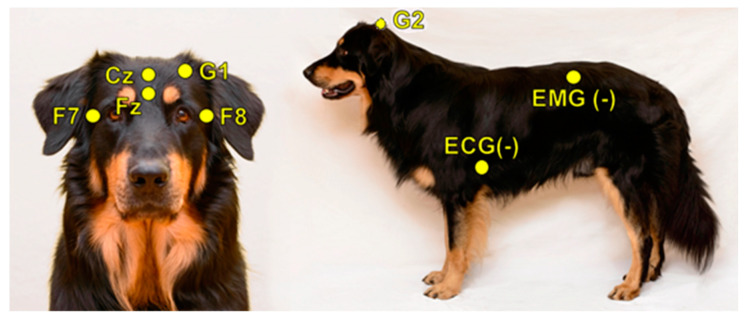
Placement of the electrodes.

**Figure 2 animals-10-00927-f002:**
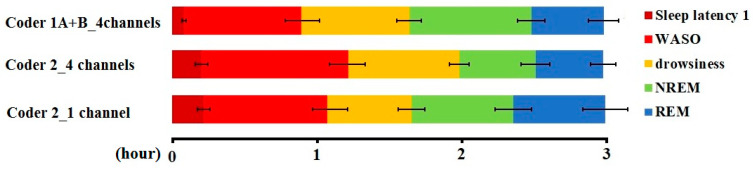
Sleep macrostructure distribution according to different coders and visible channels (mean ± SE).

**Figure 3 animals-10-00927-f003:**
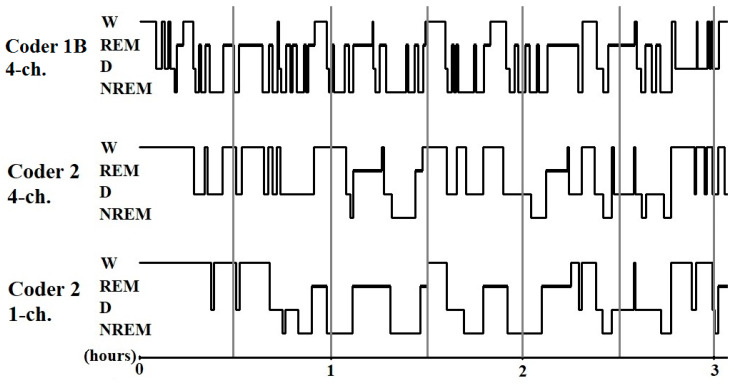
Sample hypnograms from one subject to illustrate the similarities and differences between and within coders. One typical difference between Coder 1B and Coder 2 is that Coder 1B scored the first D epoch much sooner than Coder 2. On the contrary, later epochs scored as D by Coder 1B are frequently scored as W by Coder 2. Disagreements in scoring NREM-REM/NREM-wake/NREM-D transitions were also typical.

**Figure 4 animals-10-00927-f004:**
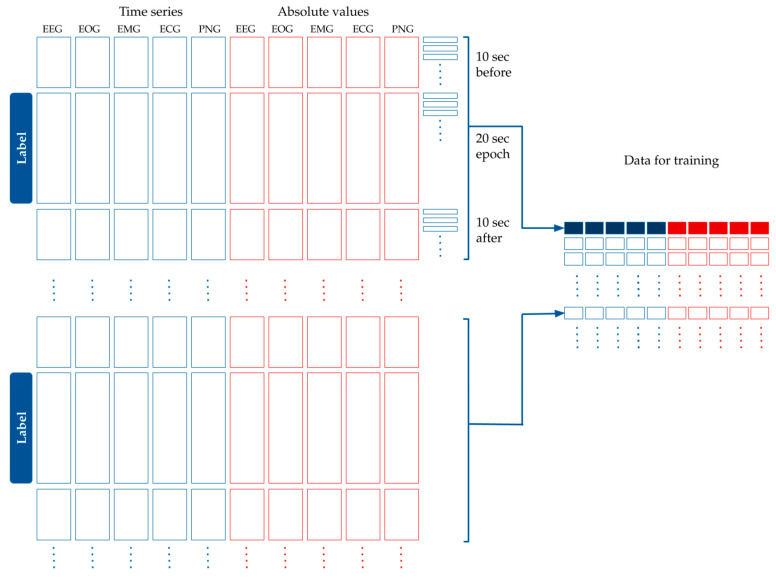
Processing the time series data for modeling. The original dataset containing the EEG, EOG, EMG, ECG and PNG time series was standardized and averaged per 100 milliseconds. The absolute values of these curves were added to the dataset. Each epoch label belongs to 40 s-long time window: a 10 s long interval before the epoch, the epoch itself, and 10 s after the epoch. Time is shown in the vertical axis.

**Figure 5 animals-10-00927-f005:**
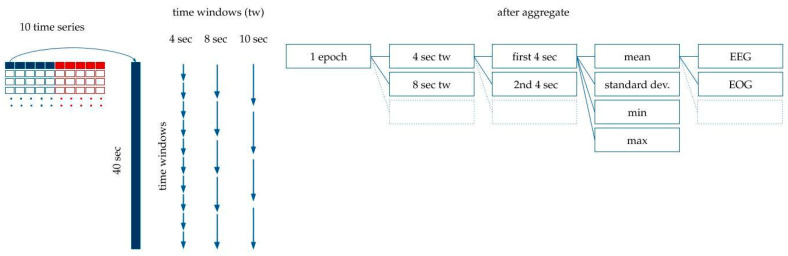
Feature generation. The 40 s-long interval was divided into subintervals with 1 s, 2 s, 3 s… 10 s, 20 s and 40 s time windows. The mean, standard deviation, and minimum and maximum values were calculated for each time window.

**Table 1 animals-10-00927-t001:** Description of the applied scoring criteria and characteristic polysomnographic view of sleep stages (EEG channel = Fz, F7, F8, Cz derivations; EOG (electrooculography) channel = F7−F8 derivations).

Sleep Stage	Coding Criteria
wake (W)	Increased high-frequency (alpha: 8.75–12.75 Hz and beta: 15–30 Hz) and fast activity in the EEG channel, high amplitude and frequency eye movements in the EOG channel, elevated muscle tone and frequent movements 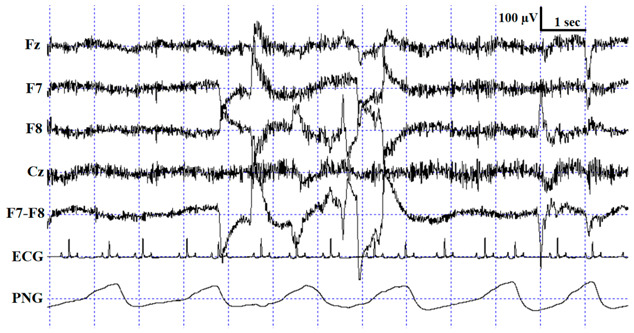
drowsiness (D)	Increased high-frequency (alpha: 8.75–12.25 Hz and beta: 12.75–30 Hz) and fast activity in the EEG channel, decreased amplitude and frequency of eye movements in the EOG channel, lowered but observable muscle tone, fairly regular respiration (PNG channel) 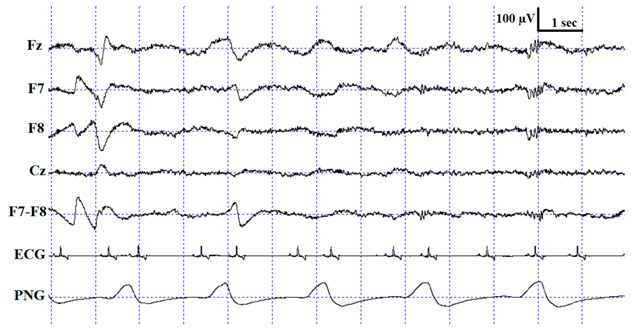
slow-wave sleep/non-REM (SWS)	Occurrence of ≥15 μV delta (1–4 Hz) activity and/or sleep spindles (waves with 12–16 Hz frequency and ≥0.5 s duration) in the EEG channel, no or low amplitude eye movements in the EOG channel, relatively regular respiration (PNG channel) and decreased muscle tone 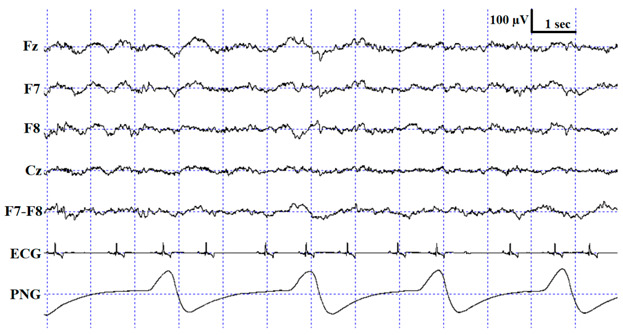
rapid eye movement (REM)	Increased theta (4.25–4.5 and 7–8 Hz) activity and fast activity in the EEG channel, occurrence of rapid eye movements in the EOG channel—also seen as artifacts in the EEG channel, muscular atonia, irregular respiration (PNG channel) and heart beat (ECG channel) 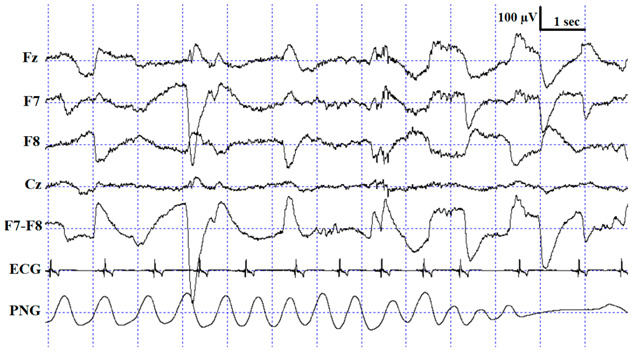

**Table 2 animals-10-00927-t002:** Values of the Cohen’s kappa (κ) in different conditions.

κ; %	Coder 1B Versus Coder 2 (4 ch.)	Coder 1 (A + B) Versus Coder 2 (4 ch.)	Coder 1A Versus Coder 2 (4 ch.)	Coder 2 1 ch. Versus Coder 2 (4 ch.)
Whole dataset	0.29; 47%	0.37; 53%	0.49; 63%	0.58; 69%
Transitions cut out (1849 epoch)		0.38; 54%		0.64; 74%
Random 1849 epochs cut out		0.38; 54%		
Random 100 epochs cut out				0.73; 81%

**Table 3 animals-10-00927-t003:** Confusion matrix for sleep stages in the within-coder condition. The darkest background color indicates the highest ratio in a row (1–0.5) and a middle deep color indicates 0.49–0.1 while the brightest indicates a 0.09–0 ratio. NREM: non-rapid eye movement; REM: rapid eye movement.

Number of Visible Channels	Stage	1 Channel			
		W	D	NREM	REM
4 channels	W	0.801	0.094	0.021	0.154
	D	0.128	0.547	0.204	0.173
	NREM	0.015	0.082	0.703	0.086
	REM	0.035	0.026	0.164	0.591

**Table 4 animals-10-00927-t004:** Confusion matrix for sleep stages between Coder 1A and Coder 2. The darkest background color indicates the highest ratio in a row (1–0.36) and a middle deep color indicates 0.35–0.1 while the brightest indicates a 0.09–0 ratio.

Coder	Stage	Coder 2			
		W	D	NREM	REM
Coder 1A	W	0.764	0.053	0.033	0.114
	D	0.280	0.515	0.106	0.179
	NREM	0.007	0.241	0.692	0.214
	REM	0.044	0.086	0.109	0.367

**Table 5 animals-10-00927-t005:** Confusion matrix for sleep stages between Coder 1B and Coder 2. The darkest background color indicates the highest ratio in a row (1–0.33) and the middle deep color indicates 0.32–0.1 while the brightest indicates a 0.09–0 ratio.

Coder	Stage	Coder 2			
		W	D	NREM	REM
Coder 1B	W	0.633	0.092	0.013	0.131
	D	0.358	0.286	0.02	0.127
	NREM	0.072	0.390	0.525	0.148
	REM	0.211	0.289	0.11	0.332

**Table 6 animals-10-00927-t006:** Results of paired t-tests on the macrostructural variables (adjusted *p* = 0.006).

Variables	Within-Coder (1 vs. 4 Channel Scoring)	Between-Coders (Coder 1A + B vs. Coder 2)
Sleep latency 1 (first D sleep, min)	t_9_ = −0.76, *p* = 0.47	t_9_ = 2.83, *p* = 0.02
Sleep latency 2 (first NREM sleep, min)	t_9_ = 0.72, *p* = 0.49	t_9_ = 2.76, *p* = 0.07
Relative W duration (%)	t_9_ = 1.37, *p* = 0.20	t_9_ = 2.1, *p* = 0.02
Relative D duration (%)	t_9_ = 1.28, *p* = 0.23	t_9_ = 1.03, *p* = 0.33
Relative NREM duration (%)	t_9_ = −1.01, *p* = 0.34	t_9_ = −2.41, *p* = 0.04
Relative REM duration (%)	t_9_ = 0.69, *p* = 0.51	t_9_ = 0.36, *p* = 0.72
WASO (W after first sleep, min)	t_9_ = 1.42, *p* = 0.19	t_9_ = 1.58, *p* = 0.15
Average sleep cycle duration (%)	t_9_ = 1.38, *p* = 0.20	t_9_ = 1.1, *p* = 0.30

**Table 7 animals-10-00927-t007:** Results of paired t-tests on the relative spectral power of the Fz channel in the within- and between-coder conditions (adjusted *p* = 0.006).

Frequency Ranges	Within-Coder			Between-Coders		
	D	NREM	REM	D	NREM	REM
Delta (1–4 Hz)	t_9_ = 1.48, *p* = 0.17	t_9_ = 0.27, *p* = 0.8	t_9_ = −0.81, *p* = 0.44	t_9_ = −1.66, *p* = 0.13	t_9_ = −1.14, *p* = 0.28	t_9_ = −0.05, *p* = 0.96
Theta (4–8 Hz)	t_9_ = −0.98, *p* = 0.35	t_9_ = 0.83, *p* = 0.43	t_9_ = 1.33, *p* = 0.23	t_9_ = 0.51, *p* = 0.62	t_9_ = 0.15, *p* = 0.89	t_9_ = −0.65, *p* = 0.53
Alpha (8–12 Hz)	t_9_ = −1.33, *p* = 0.22	t_9_ = 1.5, *p* = 0.17	t_9_ = 0.03, *p* = 0.98	t_9_ = 1.44, *p* = 0.18	t_9_ = 0.08, *p* = 0.94	t_9_ = −0.5, *p* = 0.63
Beta (12–30 Hz)	t_9_ = −1.6, *p* = 0.14	t_9_ = −1.28, *p* = 0.24	t_9_ = 0.5, *p* = 0.63	t_9_ = 2.56, *p* = 0.03	t_9_ = 2.1, *p* = 0.07	t_9_ = 1.01, *p* = 0.34

**Table 8 animals-10-00927-t008:** The top 5 features based on importance calculated as the mean of the coefficients for the given resolution.

Time Series	Aggregate Type	Curve Type	Time Window	Importance
EEG	standard deviation	absolute	40	0.583
PNG	standard deviation	absolute	40	0.469
PNG	standard deviation	absolute	20	0.409
PNG	standard deviation	absolute	10	0.339
ECG	mean	absolute	40	0.322

**Table 9 animals-10-00927-t009:** The top 5 features based on importance calculated for the GBT as the mean of the coefficients for the given time window.

Time Series	Aggregate Type	Curve Type	Time Window	Importance
EEG	mean	absolute	40	0.037
EOG	maximum	original	40	0.029
EEG	maximum	original	40	0.029
EOG	mean	absolute	40	0.026
EEG	maximum	absolute	20	0.020

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
