# Peer review of "Reliability of Family Dogs’ Sleep Structure Scoring Based on Manual and Automated Sleep Stage Identification"

_animals, 2020, doi:10.3390/ani10060927_

Round 1
Reviewer 1 Report
Well-written and thorough analytic manuscript.
Many of the technical aspects of this manuscript are beyond the scope of my knowledge, and are difficult for me to comment on. However overall this is an impressive piece of work, and while much of the technical detail was complicated, the authors did a good job of presenting the data in a logical manner. There are a few improvements the authors could make to the manuscript, mainly regarding Study 1.
First, there should be an explanation of why the number of dogs/studies were selected for analysis. It is unclear why this number was selected, whether this was done prospectively, and why certain coders only coded selected studies. The rationale needs to be more clearly explained for this section. It also seems somewhat confounding to compare intra-rater coding for 1 channel vs 4 channel rather than over the same type of recording (though this could be shown in addition, there is no baseline for understanding what intra-rater reliability would be). Second, the relevance of the findings from Study 1 could be more fully characterized. What is the impact of these findings on other researchers in the field, and what recommendations do the authors make for going forward (for interpreting findings where most codings are performed by one person, or about training in order to be able to code for studies of sleep in dogs). The authors begin to explain this, but the manuscript would be improved by further detail in the discussion section regarding the impacts of their findings on studies of sleep in dogs.
One very minor comment is that data should be plural (line 330) and generally sentences that begin with percentages are written out (Eighty-percent; line 329). In general this manuscript presents important findings that should be considered when conducting or evaluating this type of research.
Author Response
Dear Reviewer 1
Thank you for your assessment of our manuscript and for raising valid and important points. We hope that as a result of the implemented changes the paper has improved significantly and our ms will be acceptable for publication. Please find below the detailed answers to your comments.
First, there should be an explanation of why the number of dogs/studies were selected for analysis. It is unclear why this number was selected, whether this was done prospectively, and why certain coders only coded selected studies. The rationale needs to be more clearly explained for this section.
Canine EEG studies were usually conducted on 7-16 dogs, thus we decided to select 10 recordings randomly which we believe is a representative sample size for the present study. Since manual scoring is very time consuming it was feasible to choose recordings which were already scored by one of our colleagues therefore only one additional scorer had to code them for inter-observer reliability investigation. Since all of the coders of the present study completed very similar education in canine PSG scoring we consider all coders as experts who are already scored PSG recording included in several peer-reviewed articles (all cited in the ms.). We included this in the revised ms. (line 171)
It also seems somewhat confounding to compare intra-rater coding for 1 channel vs 4 channel rather than over the same type of recording (though this could be shown in addition, there is no baseline for understanding what intra-rater reliability would be).
Previously published canine EEG studies calculated inter-observer agreement on the very same recordings with same number of visible EEG channels. Thus, we already have some knowledge on the value of these types of agreements. Additionally we are planning to use more than one active EEG channel in our future studies and also to analyse bigger samples comparatively (i.e. based on 100 or more PSG recordings of dogs). For such comparisons we need to know whether 1- and multiple channels-recordings are potentially similar or not when it comes to visible scoring. The present study highlighted the importance of visible EEG channels and warn researchers to take this feature into account when they aim to compare such recordings.
Second, the relevance of the findings from Study 1 could be more fully characterized. What is the impact of these findings on other researchers in the field, and what recommendations do the authors make for going forward (for interpreting findings where most codings are performed by one person, or about training in order to be able to code for studies of sleep in dogs). The authors begin to explain this, but the manuscript would be improved by further detail in the discussion section regarding the impacts of their findings on studies of sleep in dogs.
We elaborated on this issue in the discussion section (line 475)
One very minor comment is that data should be plural (line 330) and generally sentences that begin with percentages are written out (Eighty-percent; line 329). In general this manuscript presents important findings that should be considered when conducting or evaluating this type of research.
These have been corrected in the revised ms.
Reviewer 2 Report
Brief summary: The paper entitled “Reliability of family dogs’ sleep structure scoring based on manual and automated sleep stage identification” addresses an unsolved problem: how to score canine sleep based on polysomnographic recordings in the absence of standardised criteria. Although there are functional parallels between the sleep physiology (e.g. sleep macro- and microstructure, EEG-spectra), and the role of undisturbed sleep on emotion processes and memory consolidation in humans as well as in dogs, the validity and reliability of sleep staging is still under discussion.
As a step towards standardisation and harmonisation of canine sleep scoring the authors present two studies. The first study analyses the comparisons of visual sleep scorings by three human experts and a second study presents the results of three different methods of automated sleep analyses using supervised machine learning algorithms (logistic regression models, gradient boosted trees and convolutional neural networks).
Visual scorings of human experts showed only a moderate to substantial inter-rater comparison (based on Cohen`s Kappa), with the lowest agreements in longer PSG-files (3 hours) as compared to short recordings (20 epochs ~ 4 minutes) and for transient sleep stages (e.g. sleep onset, drowsiness, transition between sleep stages). Moreover, the accordance between human scorers depended on the number of signal traces presented (one or more EEG leads) and was highly determined by their prior scoring experience. Nevertheless, the lack of disagreements between the human scorers had “no or only moderate differences in macrostructural and spectral variables”, as the authors summarised the results in the abstract (line 44, 45).
Sleep staging based on a combination of the tested three different supervised machine learning algorithms revealed better results than individual models with a final AUC > 0.8!
General comments:
1. Introduction: AASM-sleep scoring rules should be mentioned and discussed.
In a brief summary the authors reviewed the ongoing discussion of human visual sleep scoring, addressing the limitation and pitfalls (lack of interrater reliability, signal artefacts, methodological issues related to poor standardization of scoring rules). Nevertheless, the authors only reviewed articles related to the scoring rules by Rechtschaffen & Kales (R&K, 1967), ignoring that in 2007 the R&K-criteria were replaced by new criteria of the American Academy of Sleep Medicine (Iber et al. 2007). Meanwhile, the AASM-criteria were reviewed and revised several times to guarantee evidence based and state-of-the-art knowledge. Unfortunately, these developments are also not reflected by the authors, all the more since the AASM-criteria may offer some rules, which might be helpful in canine sleep scoring (e.g. merging deep sleep stages 3 and 4 into N3, alternative electrode placements and additional EEG traces, rules for scoring arousals). Moreover, the AASM rules are - in certain aspects – more suitable for automatic sleep classifier.
Study 1.
2.1. Material & Methods
Lack of consensus scorings
The presented study is based on polysomnographic sleep recordings of 10 different dogs, visually annotated by 3 independent scorers. Nevertheless, only one expert scored all recordings, the two other scorers analysed only subsamples of the 10 recordings (6 and 4 recordings). Complete scorings of all 10 recordings by 3 independent scorers and consensus finding in case of inconsistency would have resulted in a very powerful dataset and a solid “ground truth”. Unfortunately, such dataset is missing and this shortcomming weakens quantitative data analyses, especially those utilizing supervised machine learning algorithms.
Scoring criteria for W, D, REM, NREM?
According to human sleep scoring, “wake-”, “REM-” and “NREM-states” are described by frequency- and amplitude-criteria as well as by special features or patterns (wake: alpha-waves; NREM: presence of sleep spindles and K-complexes, theta-, delta-waves. 75µV-criterium; REM: beta activity, absence of muscle tone, rapid eye movements etc.). Here, references or a brief description (or table) of the applied scoring criteria would be helpful (please add this to lines 164ff).
3. Study 2
In my opinion, the second part of the paper does not really fit to the first part. For sure, there are similarities between the first and the second study: canine PSG-data (obvious other datasets than in the first part) - visual scorings - the problem of non-existence of scoring rules. Therefore, I suggest that the authors should clarify at the beginning of paragraph 3 their viewpoint why there is a need of automated sleep staging in canine sleep studies. For instance:
- replacement of the human scorer (in this case unsupervised learning algorithms may have some advantages);
- simulating human sleep scoring by machines (with this in mind, the LogReg-method is not sufficient, because EOG- and EMG-signals do not contribute - as by a human scorer - to the output);
- a semi-automatic procedure which needs the human expert for quality assurance etc.
Otherwise, the rationale why LogReg, GBT or CNN routines were chosen is not clear. Moreover: in the absence of accepted rules and standards for canine sleep scoring, the use of unsupervised NN-classifier may have some advantages over supervised algorithms.
4. General discussion
Instead of distinct sleep stages there are many attempts in human sleep research to describe sleep as a continuum by quantitative parameters (amplitude-, frequency-mapping) or by feature-plots (arousals, slow-wave activity, spindle-plots etc.). This should briefly be adressed in the discussion (especially arousals as a very important indicator of sleep-stability/-continuity).
Specific comments, suggestion:
Line 19: “untrained patients”: I would prefer “untrained individuals”
Line 81 to 83: Please note: Most artefacts are successfully eliminated or reduced by simple band pass filter, template matching / moving windows algorithms (e.g. ECG-, 50/60 Hz- or eye movements-artefacts). These techniques are mostly sufficient for visual or “manual” sleep scoring. More sophisticated techniques such as ICA are only necessary for quantitative EEG-signal analysis. Moreover, visual EEG sleep scoring may profit from artefacts since they contain valuable (additional) information: e.g. muscle artefacts indicate changes of sleep posture and sleep stages or the beginning/end of REM sleep may also cause movements or an increase of the muscle tone.
Line 165: D sleep - please indicate D (= drowsiness)
Line 255: Here, the authors use the term Stage N1 (which refers to the AASM sleep scoring rules 2007, but not to Rechtschaffen & Kales). The cited literature (14,16,17,24) was published before 2007, when the term S1 was in use. Therefore, please replace "Stage N1" by "Stage S1/(N1)".
Line 314, 315: “for machine learning expert annotations from 6 different scorers were used”. Here, the description of the used data sets is too vague: did the authors use any additional selection criteria (e.g. exclude all epochs with ambiguous data or transient sleep stages)?
Line 412: delete the first bracket ((e.g. arguments ....)
Line 415: "While sleep spindle identification is a task that divides human researchers ...". Please review this sentence: does it mean that the use of automated sleep spindle identification is discussed controversially?
Author Response
Dear Reviewer 2
Thank you for your assessment of our manuscript and for raising valid and important points. We hope that as a result of the implemented changes the paper has improved significantly and our ms will be acceptable for publication. Please find below the detailed answers to your comments.
- Introduction: AASM-sleep scoring rules should be mentioned and discussed.
In a brief summary the authors reviewed the ongoing discussion of human visual sleep scoring, addressing the limitation and pitfalls (lack of interrater reliability, signal artefacts, methodological issues related to poor standardization of scoring rules). Nevertheless, the authors only reviewed articles related to the scoring rules by Rechtschaffen & Kales (R&K, 1967), ignoring that in 2007 the R&K-criteria were replaced by new criteria of the American Academy of Sleep Medicine (Iber et al. 2007). Meanwhile, the AASM-criteria were reviewed and revised several times to guarantee evidence based and state-of-the-art knowledge. Unfortunately, these developments are also not reflected by the authors, all the more since the AASM-criteria may offer some rules, which might be helpful in canine sleep scoring (e.g. merging deep sleep stages 3 and 4 into N3, alternative electrode placements and additional EEG traces, rules for scoring arousals). Moreover, the AASM rules are - in certain aspects – more suitable for automatic sleep classifier.
Our canine hypnogram scoring was mainly based on the R&K rules, but based on this suggestion we now included a section about the AASM scoring method and its potential role in future canine sleep research in the discussion section (line 513).
Study 1.
2.1. Material & Methods
Lack of consensus scorings
The presented study is based on polysomnographic sleep recordings of 10 different dogs, visually annotated by 3 independent scorers. Nevertheless, only one expert scored all recordings, the two other scorers analysed only subsamples of the 10 recordings (6 and 4 recordings). Complete scorings of all 10 recordings by 3 independent scorers and consensus finding in case of inconsistency would have resulted in a very powerful dataset and a solid “ground truth”. Unfortunately, such dataset is missing and this shortcomming weakens quantitative data analyses, especially those utilizing supervised machine learning algorithms.
Our Canine PSG database contains sleep recordings scored by a single researcher each. Thus when starting this study we had to choose between using recordings scored by the same scorer (that would result in ten recordings scored by the same two researchers) or recordings scored by different scorer (this could have resulted in ten recordings with several different “first” coders and the same second coder). We decided to use recordings of two “first” coders and of different experience in order to be able to tell something about the effect of coder identity. We, however, fully agree with the notion that scoring all ten (or more) PSG recordings by three (to several) coders would result in an even stronger dataset. At the same time, we believe the results of the present study are valuable and contain not only the first ever such results, but also important findings relating to several aspects of canine hypnogram scoring (e.g. the number of visible channels).
Scoring criteria for W, D, REM, NREM?
According to human sleep scoring, “wake-”, “REM-” and “NREM-states” are described by frequency- and amplitude-criteria as well as by special features or patterns (wake: alpha-waves; NREM: presence of sleep spindles and K-complexes, theta-, delta-waves. 75µV-criterium; REM: beta activity, absence of muscle tone, rapid eye movements etc.). Here, references or a brief description (or table) of the applied scoring criteria would be helpful (please add this to lines 164ff).
We added a Table with detailed scoring criteria and representative polysomnographic view of the different stages (see Table 1).
- Study 2
In my opinion, the second part of the paper does not really fit to the first part. For sure, there are similarities between the first and the second study: canine PSG-data (obvious other datasets than in the first part) - visual scorings - the problem of non-existence of scoring rules. Therefore, I suggest that the authors should clarify at the beginning of paragraph 3 their viewpoint why there is a need of automated sleep staging in canine sleep studies. For instance:
- replacement of the human scorer (in this case unsupervised learning algorithms may have some advantages);
- simulating human sleep scoring by machines (with this in mind, the LogReg-method is not sufficient, because EOG- and EMG-signals do not contribute - as by a human scorer - to the output);
- a semi-automatic procedure which needs the human expert for quality assurance etc.
Otherwise, the rationale why LogReg, GBT or CNN routines were chosen is not clear. Moreover: in the absence of accepted rules and standards for canine sleep scoring, the use of unsupervised NN-classifier may have some advantages over supervised algorithms.
We understand that the justification of the ML method remains unclear. In the current, updated manuscript we updated Paragraph 3 and clearly stated the importance and the need for automatic sleep scoring.
The rationale behind using LogReg and GBT models in the first place is twofold. First, we selected the LogReg classifier as a baseline since with the right feature selection linear models are capable of delivering strong baseline performance. Then, in order to improve performance, we implemented GBT since this supervised, non-linear, tree based model is generally robust to noise or overfitting and has shown superior performance compared to many other supervised algorithms recently (Chen & Guestin, 2016). Second, both LogReg and GBT have the advantage that they give us feature importance rankings providing insight into the data and model.
Finally, note that both the LogReg and GBT models work with feature generation. We applied the CNNs to explore the possibility of creating classifiers without the need of feature generation from the data.
We do not claim that these classifiers are the best but wanted to highlight the applicability of supervised ML models on the task of automatic sleep scoring. Since these methods result in sufficient performance we believe automatic scoring is indeed feasible. Finally, we thank the suggestion of NN models. To our best understanding NN models are also supervised and might be a good choice for further investigation.
- General discussion
Instead of distinct sleep stages there are many attempts in human sleep research to describe sleep as a continuum by quantitative parameters (amplitude-, frequency-mapping) or by feature-plots (arousals, slow-wave activity, spindle-plots etc.). This should briefly be adressed in the discussion (especially arousals as a very important indicator of sleep-stability/-continuity).
We added a brief section to the discussion in the revised ms (line 519).
Specific comments, suggestion:
Line 19: “untrained patients”: I would prefer “untrained individuals”
Corrected in the revised ms.
Line 81 to 83: Please note: Most artefacts are successfully eliminated or reduced by simple band pass filter, template matching / moving windows algorithms (e.g. ECG-, 50/60 Hz- or eye movements-artefacts). These techniques are mostly sufficient for visual or “manual” sleep scoring. More sophisticated techniques such as ICA are only necessary for quantitative EEG-signal analysis. Moreover, visual EEG sleep scoring may profit from artefacts since they contain valuable (additional) information: e.g. muscle artefacts indicate changes of sleep posture and sleep stages or the beginning/end of REM sleep may also cause movements or an increase of the muscle tone.
We revised this paragraph to present the corresponding information in a more efficient way.
Line 165: D sleep - please indicate D (= drowsiness)
Corrected in the revised ms.
Line 255: Here, the authors use the term Stage N1 (which refers to the AASM sleep scoring rules 2007, but not to Rechtschaffen & Kales). The cited literature (14,16,17,24) was published before 2007, when the term S1 was in use. Therefore, please replace "Stage N1" by "Stage S1/(N1)".
Corrected in the revised ms.
Line 314, 315: “for machine learning expert annotations from 6 different scorers were used”. Here, the description of the used data sets is too vague: did the authors use any additional selection criteria (e.g. exclude all epochs with ambiguous data or transient sleep stages)?
We only excluded recordings with minimal or no sleep duration (the proportion of sleep labels should be more than 5%, the proportion of awake labels at least 2%). We excluded three recordings. Other than that, we included each recording without further modification either as a train or test sample in our measurements. We clarified this in the updated manuscript.
Line 412: delete the first bracket ((e.g. arguments ....)
Corrected in the revised ms.
Line 415: "While sleep spindle identification is a task that divides human researchers ...". Please review this sentence: does it mean that the use of automated sleep spindle identification is discussed controversially?
We have clarified this sentence. We meant to say here that there are arguments both for and against automated versus manual sleep spindle detection.
Reviewer 3 Report
This manuscript describes two studies in which a large number of dogs' polysomnography recordings were analysed, in order to compare the impact of various factors (e.g. part of the recordings analysed, expertise of the raters). It also attempts to create a semi-automated procedure using a computer algorithm. This is a good piece of work that I recommend for publication in Animals, but I have some suggestions for improving clarity. In many of these cases, the terms are defined in the text, but not where they are first mentioned (eg, NREM, WASO).
L17 suggest adding 'in pets ' after '...certain behavioural problems...'
L19 suggest adding 'veterinary' after 'untrained'
L24-25 are clear after reading the ms, but not when reading the summary as a standalone
L27 - briefly define EEG spectrum, or rephrase, since this is meant to be a lay summary
L42 - add length of the epochs (ie 20sec)
L43 - write out NREM as non-REM
L64 - remove 'inarguably'
L67 - golden should be gold
L76 - explain what is meant by 'feature generation
L94 - write out NREM as non-REM
L99 - define WASO
L165 - write length of epochs
L239 - should table 1a and 1b be table 2a and 2b?
L252 - reliably should be reliability
L253 '...effort has been expanded of...' is awkward. Please rephrase
L275 - suggest removing 'see also figure 3'. Fig 3 doesn't compare humans and dogs
L279-280 change to '...Wake and NREM durations did not differ after correcting for multiple comparisons'
L282 - add corrected alpha-level for reference
Table 4 - one of the cells is highlighted. Why? it's not sig with the adjusted alpha level
L315 - '304 recordings of ca. 3hour duration...' add 'each' for clarity
L461 - sentence fragment
L473 - your own results suggest the same, isn't that correct?
Author Response
Dear Reviewer 3
Thank you for your assessment of our manuscript and for raising valid and important points. We hope that as a result of the implemented changes the paper has improved significantly and our ms will be acceptable for publication. Please find below the detailed answers to your comments.
L17 suggest adding 'in pets ' after '...certain behavioural problems...'
L19 suggest adding 'veterinary' after 'untrained'
L24-25 are clear after reading the ms, but not when reading the summary as a standalone
This sentence was broken up into two to make the standalone meaning clear.
L27 - briefly define EEG spectrum, or rephrase, since this is meant to be a lay summary
L42 - add length of the epochs (ie 20sec)
L43 - write out NREM as non-REM
L64 - remove 'inarguably'
L67 - golden should be gold
L76 - explain what is meant by 'feature generation
L94 - write out NREM as non-REM
L99 - define WASO
L165 - write length of epochs
L239 - should table 1a and 1b be table 2a and 2b?
L252 - reliably should be reliability
L253 '...effort has been expanded of...' is awkward. Please rephrase
L275 - suggest removing 'see also figure 3'. Fig 3 doesn't compare humans and dogs
L279-280 change to '...Wake and NREM durations did not differ after correcting for multiple comparisons'
L282 - add corrected alpha-level for reference
Table 4 - one of the cells is highlighted. Why? it's not sig with the adjusted alpha level
L315 - '304 recordings of ca. 3hour duration...' add 'each' for clarity
L461 - sentence fragment
Corrections according to L17-L461 were amended in the revised ms.
L473 - your own results suggest the same, isn't that correct?
We think that our results are contradictory in this aspect. Relatively low inter-observer agreement values in the present study seem to support this suggestion, however the fact that macrostructural and spectral results were similar between coders highlighted the efficiency of this requirement.